# Putting Cells in Motion: Advantages of Endogenous Boosting of BDNF Production

**DOI:** 10.3390/cells10010183

**Published:** 2021-01-18

**Authors:** Elvira Brattico, Leonardo Bonetti, Gabriella Ferretti, Peter Vuust, Carmela Matrone

**Affiliations:** 1Center for Music in the Brain, Department of Clinical Medicine, Aarhus University & The Royal Academy of Music Aarhus/Aalborg, 8000 Aarhus, Denmark; leonardo.bonetti@clin.au.dk (L.B.); Vuust@clin.au.dk (P.V.); 2Department of Education, Psychology, Communication, University of Bari “Aldo Moro”, 70121 Bari, Italy; 3Unit of Pharmacology, Department of Neuroscience, Faculty of Medicine, University of Naples Federico II, via Pansini 5, 80131 Naples, Italy; gabriella.ferretti@unina.it

**Keywords:** music, BDNF, BDNF gene

## Abstract

Motor exercise, such as sport or musical activities, helps with a plethora of diseases by modulating brain functions in neocortical and subcortical regions, resulting in behavioural changes related to mood regulation, well-being, memory, and even cognitive preservation in aging and neurodegenerative diseases. Although evidence is accumulating on the systemic neural mechanisms mediating these brain effects, the specific mechanisms by which exercise acts upon the cellular level are still under investigation. This is particularly the case for music training, a much less studied instance of motor exercise than sport. With regards to sport, consistent neurobiological research has focused on the brain-derived neurotrophic factor (BDNF), an essential player in the central nervous system. BDNF stimulates the growth and differentiation of neurons and synapses. It thrives in the hippocampus, the cortex, and the basal forebrain, which are the areas vital for memory, learning, and higher cognitive functions. Animal models and neurocognitive experiments on human athletes converge in demonstrating that physical exercise reliably boosts BDNF levels. In this review, we highlight comparable early findings obtained with animal models and elderly humans exposed to musical stimulation, showing how perceptual exposure to music might affect BDNF release, similar to what has been observed for sport. We subsequently propose a novel hypothesis that relates the neuroplastic changes in the human brains after musical training to genetically- and exercise-driven BDNF levels.

## 1. Introduction

Brain-derived neurotrophic factor (BDNF) is a key molecule involved in neuronal plastic changes related to learning and memory. Adequate BDNF levels are crucial in both normal and pathological aging, as well as in psychiatric diseases, in particular in brain areas that are involved in the codification of memory processes, such as the hippocampus and parahippocampal areas [1,2,3,4].

Several lines of evidence from animal models and physiological studies with humans point to the idea that motor exercise reliably boosts BDNF levels [5]. Furthermore, it has been suggested that these molecular effects are linked to systemic changes in neocortical and subcortical brain regions [6]; these might be the mechanisms mediating the advantage of motor exercise on a plethora of diseases. Indeed, neurocognitive studies on humans link the modified brain functions after motor exercise to behavioural changes related to mood regulation, memory, well-being, and even cognitive preservation in aging and neurodegenerative diseases [7,8].

These effects have been seen in relation to whole-body exercise, such as running on a wheel for animal models [9,10] or sport in humans [5,11,12]. However, the effects of a special kind of motor exercise, involving music, have only been considered in animal models (Table 1). Most trivially, playing a musical instrument requires moving the limbs and refining motor actions and programs, as well as mapping sounds to actions and back [13]. This special motor activity leads to measurable modifications of the dedicated neural circuity, with superior functionality and connectivity [14,15,16,17,18], as well as increased thickness and volume of grey and white matter of audio-motor areas in the cerebrum and cerebellum [19,20], as visible in both cross-sectional studies comparing professional musicians (dedicating decades of their life and tens of thousands of hours to playing a musical instrument) [21] and controls and in longitudinal studies following the learning process in children or adults [22,23].

Motor exercise occurs not only when playing a musical instrument and moving the limbs, but also when listening to music. When we think of an upbeat pop song, we immediately and almost involuntarily want to move, bouncing our head, tapping our feet, or even dancing with our whole body. Such response could be congenital, being observed in young children [24], and seems specifically human or at least restricted to vocal learning species, being seen only in rare instances such as in parrots or sea lions, apart from humans [25]. In addition to that, when we learn a song well or when we learn to play an instrument, the action observation brain network activates in humans [26], and also in primates [27]. In this commentary, we argue for the relevance of music for neurotrophic-mediated processes. Hence, after reviewing relevant literature on sport exercise and the putative cellular and molecular mechanisms for its beneficial effects, we call for further research towards testing the hypothesis that the beneficial effects of music for brain health in terms of neuroplasticity and neural preservation would be mediated by music-induced neurotrophin release.

## 2. BDNF: Gene Structure and Protein Localization

BDNF belongs to a class of neurotrophic factors sharing structural similarities and regulating multiple and common biologic processes, such as neuronal development, differentiation, and apoptosis [28]. In particular, BDNF is the second member of the neurotrophic factor family that has been identified, after neuronal growth factor (NGF) and before neurotrophin-3 and neurotrophin-4/5 [29,30].

Indeed, BDNF, NGF, neurotrophin-3, and neurotrophin-4/5 have a crucial role in proliferation, differentiation, and survival of neuronal populations during development and participate in a variety of learning and memory functions [28,30].

The complex structural organization of the *Bdnf* gene underlines the multifaceted roles exerted by this protein in the central and peripheral nervous system. Human, rat, and mouse BDNF are expressed from a single gene locus. The human *Bdnf* gene is located on chromosome 11p13, and has 11 exons and 9 functional promoters, all producing at least 34 different transcripts existing in two isoforms, either with long or short 3′ UTR [31,32].

Each of these transcripts can be differently expressed in various tissues. *Bdnf* gene transcription is tightly regulated, cell-type specific, and controlled by neural activity. In particular, transcripts containing exons II and VII are expressed exclusively in the brain, transcripts containing exons II, III, IV, V, and VII are expressed predominantly in the brain but also in certain peripheral tissues, and transcripts containing exons VI and IXabcd show a wide pattern of expression [31,33]. This complex *Bdnf* gene structure is believed to be essential for BDNF regulation at different levels through the interaction with transcription regulatory factors or mRNA-targeting signals [34,35].

Other factors, such as physical exercise, seizures, ischemia, osmotic stress, and antidepressant treatment have also been involved in regulating *Bdnf* expression levels, either at the promoter level, or by controlling BDNF translation modifications and stability [1,36]. Meanwhile, the intricate structure and regulation of *Bdnf* gene activity provide a broad susceptibility for the epigenetic control of *Bdnf* expression [37,38,39].

BDNF is synthesized in the brain, but it can be secreted either by neurons under physiological conditions or by astrocytes following injury or inflammation [36]. BDNF synthesis occurs in regions that participate in emotional and cognitive function, namely sensory cortices, hippocampus, amygdala, basal forebrain, dorsal vagal complex, hindbrain, and midbrain [40,41,42]. From these areas, BDNF can be retrogradely transported, thus reaching the cell bodies of the Raphe nuclei and locus coeruleus [43,44]. Within these brain areas, at the cellular level, BDNF is predominantly somatodendritic, but it is also present in the dendrites, and in close proximity to spines either in pre- or postsynaptic compartments [34,45]. BDNF can undergo both retrograde and anterograde transport, and this appears to be important for conferring the ability of locally translated BDNF to modulate synaptic transmission and synaptogenesis [37,46,47].

BDNF expression level changes during neuronal development: it is low during fetal development, markedly increases after birth, and then decreases in adults [3,48]. Hence, there exist two types of BDNF: pro- and mature BDNF, both present in the human body [4]. BDNF is initially synthesized as a precursor, pro-BDNF, and then cleaved into a mature BDNF by furin or proconvertase enzymes [49]. The ratio of pro-BDNF to mature BDNF is highest in the neonatal and adolescent stages in mice, whereas in adulthood, mature BDNF predominates [50].

BDNF activity depends on its binding to two receptors, TrkB and p75 [51,52]. In particular, proBDNF has higher binding affinity to p75, thus activating downstream signalling, resulting in the reduction of the spine complexity and density, induction of long-term depression (LTD), promotion of neuronal cell death in programmed cell death, and motor axon pruning of rejected connections [53,54,55]. In contrast, mature BDNF preferentially binds to TrkB receptors in an activity-dependent manner, and increases cell survival and differentiation, dendritic spine complexity, long-term potentiation (LTP), synaptic plasticity, and the resculpting of networks [52,56].

In adults, expression of the BDNF gene in the brain leads to regulation of both excitatory and inhibitory synaptic transmission and activity-dependent plasticity. Under adverse conditions, such as glutamatergic stimulation, cerebral ischemia, hypoglycaemia, and neurotoxicity, BDNF, as well as NGF, has been demonstrated to exert neuroprotective effects by activating pro-survival and antiapoptotic mechanisms [57,58,59,60,61,62]. BDNF and TkrB contribute to synaptic plasticity by aiding existing survival of existing neurons and facilitating the growth of new neurons (neurogenesis) [63,64] or new synapses (synaptogenesis) [65]. All these cellular processes can be tracked behaviourally to the formation of new memories, namely the translation of short-term memory traces to a longer-term storage through changes in the structure of nerve circuits, following genetic transcriptional processes in hippocampal areas (where BDNF is mainly expressed) [66]. These processes can be reproduced in the lab by means of rapid pairing of presynaptic activity with postsynaptic depolarization for each of the three main synaptic pathways of the hippocampus (via perforating fibers, muscoid fibers, and Schaffer collateral fibers). Studies have demonstrated that BDNF is necessary for LTP: differences in levels of BDNF influence LTP, and the blockage of BDNF binding to TrkB nullifies LTP and neurogenesis [66].

As stated earlier, after synthesis in several cerebral regions, including the entorhinal cortex, BDNF is anterogradely trafficked to the hippocampus. In the hippocampal region, the activity-dependent release of BDNF seems to have a key role in supporting the electrophysiological function of memory circuitry [62]. This is supported by translational evidence that BDNF, as well as NGF levels and signals, become deficient in the entorhinal cortex and the hippocampus in Alzheimer’s disease (AD) [67,68,69].

## 3. BDNF in Health and Disease

The complexity of the BDNF gene, with its multiple mechanisms of regulation at the transcriptional and translational levels, as well as BDNF neuronal functions at the pre- and post-synaptic level, all converge to modulate personality traits and cognitive functions, particularly learning and memory, in humans [62]. Moreover, it is known that NGF and BDNF contribute to the survival of neurons and counteract neuronal degeneration and apoptosis in in vivo and in vitro AD neuronal models [57,70].

Consistently, a decrease in BDNF levels has been reported in neurodegenerative disorders, such as AD [69,71], Huntington’s disease (HD) [72], and epilepsy [73], and recent strategies aimed at sustaining or even amplifying the production of BDNF have been proved to be beneficial in patients with HD [74], AD [75,76], Parkinson’s disease (PD) [77,78], amyotrophic lateral sclerosis (ALS) [79,80,81], stroke [82,83,84], and spinal cord injury [85,86,87,88]. However, it is worth mentioning that despite the extensive literature dating back to the early 1990s, in which reduced BDNF levels in the brain have been associated with neurodegenerative diseases, only in patients with HD has a genetic defect in BDNF gene been mechanistically related to the pathology [89]. In fact, among the neurodegenerative diseases, HD is the one in which reduced BDNF levels best correlate with the onset and progression of the pathology [72,90].

In turn, conflicting studies performed in postmortem brain tissues from individuals with AD have shown BDNF mRNA levels either decreased in the neocortex and in the Meynert nucleus basalis—where cholinergic innervation of the cerebral cortex are mostly present [91,92,93]—or increased in the hippocampus [94,95]. Contrarily, in postmortem brains of PD patients, a reduction in the transcription of the BDNF gene has been reported in the putamen striatal neurites [96], as well as in the substantia nigra pars compacta [97].

BDNF Met66 polymorphism, a common single nucleotide polymorphism of rs6265 in the BDNF gene, consists in the substitution of valine (Val) to methionine (Met) at the codon 66 (Val66Met, c.196 G > A, dbSNP: rs6265) near the middle of the BDNF pro-domain [98]. For several decades, the BDNF pro-domain has been considered a proteolytic fragment of pro-BDNF. However, clues to the potential functions of BDNF pro-domain arose from the discovery that it is expressed in the hippocampus of rodents, especially during adolescence and adulthood, and it is released from neurons upon depolarization [49]. In addition, recent studies have reported that the BDNF Met66 pro-domain variant shapes the BDNF pro-domain architecture and triggers its binding to p75 and sortilin-related Vps10p domain containing receptor 2 (SorCS2), thus causing the neuronal cone retraction and the remodeling of the neuronal morphology [98]. BDNF Met66 polymorphism also affects the activity-dependent release of the BDNF protein, thus reducing the amount of BDNF that is delivered to synapses [99]. Previous research linked the BDNF Met66 polymorphism to altered brain functionalities and psychological capacities among cognitive [100,101,102,103,104] and emotional domains [105,106].

For instance, BDNF Met66 polymorphism was recently associated with a higher risk of cognitive impairment in PD [101,103], bipolar disorder [105], and depression [106]. In contrast, a controversial involvement of the BDNF Met66 variant has been described in AD. Fukumoto et al. [107] associated the BDNF Met66 allele to an increased susceptibility to AD in women but not in men, while other studies have argued for an association of Met66 with changes in hippocampal volume and brain connectivity [108,109]. Another polymorphism in the *Bdnf* gene, C270T (rs56164415), located in a non-coding region, was found to increase the risk of late-onset AD [110]. Of interest a study carried out in a restricted group of patients with HD who were heterozygous for the BDNF Met66 polymorphism showed a later onset of HD [111,112,113]. However, subsequent studies failed to confirm this linkage [114,115].

Polymorphisms on BDNF receptors have also been reported. In particular, rs2289656 TrkB polymorphism has been associated with an increased risk of suicidal ideation in depressive patients [116], whereas the polymorphism S205L in the p75^NTR^ receptor has been suggested to be protective against suicidal behavior [116].

In addition, stress and post-traumatic stress disorder (PTSD)-like behavior is connected to lower BDNF levels [117]. Indeed, animals exposed to either acute or chronic stress have presented reduced BDNF mRNA expression compared to animals that lived in normal conditions, although a steeper reaction was noticed for the acute stress condition [118]. This reduction could play a role in the neuronal atrophy of hippocampal volume associated with chronic stress and depression, as observed in both animals and humans. Even BDNF DNA methylation is affected in a PTSD rat model, possibly leading to the persistent cognitive deficits typical of PTSD. In an experiment with adult male rats exposed to two cat encounters and 31 days of social instability, increased BDNF DNA methylation was obtained in the dorsal hippocampus, especially with regards to the dorsal CA1 subregion and decreased methylation in the ventral hippocampus (CA3) [39].

All these findings have brought scientists to propose a stress–sensitivity hypothesis for BDNF, stating that the disruption of endogenous BDNF activity by epigenetic factors increases sensitivity to stress and thus vulnerability to stress-inducible disorders [119]. According to this hypothesis, on the one hand, genetically-driven BDNF levels would promote plasticity to strengthen the encoding of fear and trauma, whereas on the other hand, they would enable adaptive plasticity during extinction learning, by aiming to remove PTSD-like fear responses.

BDNF has also been suggested as a potential protective factor for stress-induced dysfunctions in relation to LTP and spatial learning and memory [120]. In the left hippocampus of rats, BDNF was infused (0.5 μL/h) for 14 days, starting 7 days before the exposure to stressful conditions. The BDNF group appeared to be protected from the negative effects of stress, and thus provided an indistinguishable performance with respect to control animals not exposed to any stressful condition. The results of this study demonstrated that LTP, spatial learning, and memory, usually compromised in chronic stress conditions, could be protected by the administration of BDNF [120].

The deficiencies in the levels or activity of BDNF in the brain of individuals affected by the above-mentioned neurologic diseases suggest that BDNF supply is a reasonable strategy to improve mental health in these patients and improve some of their neurologic symptoms. Therefore, agents that boost BDNF might promote neurogenesis and activate downstream signalling, ameliorating neurodegenerative processes in the brain (Figure 1).

Unfortunately, BDNF, as NGF, is a highly charged protein that does not readily cross the blood–brain barrier (BBB), thus making the translation of such treatments into a clinical setting challenging. Among the 7000 drugs registered in the Comprehensive Medicinal Chemistry database, only 5% can successfully treat neurological diseases, because 100% of large-molecule pharmaceuticals (antibodies, recombinant proteins, gene therapeutics) and even some smaller ones do not cross the BBB [121]. This is the reason why approaches for the developments of compounds for reversible crossing of the BBB have received significant attention in the last four decades. Currently, there are about 6700 publications registered in PubMed containing “brain drug delivery”. Despite advanced approaches, such as stem cells or small drug molecules, including neurotrophins that bypass the BBB and enhance the delivery of neurotrophins to the CNS, having resulted in improvement in functional recovery, none of these has currently found clinical application [121]. In an attempt to increase BDNF permeability through the BBB, intranasal delivery in animal models was proposed [75,122,123]. However, the reduced permeability of the nasal mucus limits the absorption efficiency [79].

Even focused ultrasound has been proposed, and although invasive and not always effective [124], encouraging studies performed in a mouse model of PD have provided perspectives in the use of focused ultrasound for improving the delivery of intranasally administered BDNF within the brain [125,126]. In addition, therapeutic effects of the BDNF injection have been assessed in non-human primate models. BDNF gene delivery to the entorhinal cortex has significantly ameliorated lesion-induced entorhinal cortical neuronal death, improved hippocampus-dependent memory, and increased neuronal size [127,128].

Recently, an elegant behavioral study investigated the role of physical activity in a genetic mouse model of AD, likely providing the most compelling evidence of how physical activity can counteract AD memory impairment by promoting adult hippocampal neurogenesis and increasing BDNF release [129] (Figure 1).

As alternative approaches, strategies to boost endogenous BDNF or TrkB receptor activation have been explored, such as the use of drugs that promote *Bdnf* transcription, translation, and protein secretion [130]; BDNF mimetics, such as LM22A-4 [131]; TrkB-FL agonists, such as 7,8-dihydroxyflavone [132]; TrkB-FL transactivators [133]; or facilitators of TrkB-FL-mediated effects, such as adenosine A_2A_ receptor (A_2A_R) agonists [134]. Nonetheless, to the best of our knowledge, none of these compounds is approved for clinical uses.

## 4. Boosting Endogenous BDNF: Sport

As reviewed above, accumulating evidence demonstrates that BDNF levels can be modulated endogenously in various ways: stress, PTSD-like behavioral stress, or adverse early experiences decrease BDNF levels, whereas mild physical exercise seems to increase BDNF levels [119,135]. In humans, sport practice is, indeed, an effective way of boosting endogenous BDNF expression, and several studies have shown the link between BDNF regulation and the amount of exercise performed. For instance, downhill running can induce time-dependent upregulation of skeletal muscle [136], with an increase of post-exercise BDNF mRNA level 5d and 7d; BDNF protein level 1d, 3d, 7d, and 14d; and continuously augmented serum BDNF levels. Furthermore, the relationship between low-intensity, high-volume (HV) and high-intensity, low-volume (HI) resistance training was reflected in increased BDNF plasma concentrations after a seven-week, intense lifting training program [11]. Overall, athletes show higher basal BDNF levels than sedentary control participants [12]. However, when comparing different sport branches (both pre- and post-training), such as combat sports mainly based on attention and concentration skills and athletic sports requiring a high resistance during competition, combat sports are associated with a higher BDNF level than that obtained during athletic sports. These findings indicate that BDNF helps maintain high levels of concentration and attention, and possibly plays a role in brain damage protection, especially with physically traumatic combat sports (Figure 1).

Consistent with these findings in humans, a meta-analysis of 29 studies, spanning 1111 human participants, analyzed BDNF expression levels across various exercise paradigms [5]. Many of the studies predominantly examined moderate exercise, although several studies did not report intensity level. Considerable evidence from this meta-analysis suggests that humans also experience a dose–response relationship, in which each session of exercise corresponds to a dose of increased BDNF expression. Furthermore, regular exercise in moderate amounts would increase the magnitude of BDNF expression following individual sessions of exercise [5].

The results obtained with humans are further validated by animal research. For instance, in a study with adult male rats, BDNF levels were measured in a number of brain regions in response to ad libitum access to running wheels for a varied period of time, ranging from 0 to 7 nights [9]. After nights of wheel running, BDNF mRNA was significantly augmented in several regions, and especially within the hippocampus. Similarly, a single bout of acute highly intense physical exercise on the treadmill was sufficient to increment BDNF mRNA expression in sedentary mice, but did not modulate BDNF levels in mice having regular access to the running wheel for one month [10].

## 5. Boosting Endogenous BDNF: Music

The number of studies that have investigated music-related biology in healthy and diseased human brains have grown exponentially during the past decades [6]. In animal models, music (vs. noise) modulates neurogenesis and adaptation by increasing cortical and subcortical BDNF levels [137,138,139,140]. For instance, in 10 mice it has been shown that exposure to new-age music of slow rhythm and low intensity (50–60 dB) 6 h/day for 21 consecutive days (limited to the dark periods so as not to disturb mice during sleep) through a compact disc (CD) player played in the room modulates neuroplasticity by increasing BDNF levels in the hippocampus [138] and hypothalamus [139], compared to control mice placed in a similar room with ambient noise (about 50 dB). Remarkably, after the 21 days of environmental enrichment, mice showed improved learning performance in a passive avoidance task. Moreover, it has been found that exposure to music by Mozart (65–75 dB) for 7 days in utero and at least 60 days postpartum (only during dark periods) increased the levels of TrkB in the cortex of postnatal mice, as well as their performance (fewer errors) in a maze learning task, compared with mice similarly exposed to white noise (70 dB) or to ambient noise (55 dB) [140]. Supportive findings of improved BDNF/Trk levels in the dorsal hippocampus and dentate gyrus of developing rats after exposure to music by Mozart were associated with significantly better learning performance in a Morris water maze test, compared with mice that were not exposed to music [141].

These BDNF studies where mice or rats were peculiarly juxtaposed to artistic music were inspired by the so-called “Mozart effect”, first identified in humans in a *Nature* paper from 1993 [142] and then repeated with animals [143]—namely, a cognitive advantage conferred simply by listening to Mozart’s music. The original *Nature* findings reported improved performance in three spatial tasks after listening for 10 min to a Mozart piano sonata (K. 448), compared to listening to relaxation instructions or to sitting in silence. In subsequent behavioral studies with humans [144,145,146], these temporary effects of Mozart’s music were either not replicated, or were found to be related to increased arousal levels rather than to the music itself [147,148]. However, generally positive effects of long-term music training on cognitive functions have been supported by both behavioral and neuroimaging evidence [149], irrespective of the composer or the musical style practiced. For instance, after several months of learning to play a musical instrument, children have shown several improved cognitive functions [150,151,152,153], such as working memory (especially auditory), inhibitory control, and spatial and logical intelligence. Moreover, professional musicians seem to have higher intelligence quotient (IQ) scores than controls, when all other variables are matched, and in a positive correlation with years of training [154].

BDNF affects both the neurogenesis and later adaptation of the auditory system, especially with prenatal exposure to music [155]. In relation to this, multiple studies have demonstrated that prenatal exposure to music was responsible for increasing the BDNF levels within the hippocampus of chicks [156]. Moreover, exposure to music in the juvenile age results in a notable increase in BDNF expression that can protect in later stressful events, thus preventing anxiety-like behaviors [137]. Hence, music has been suggested as an effective contribution to therapeutic intervention for psychiatry disorders, thanks to its regulation of BDNF levels, which are of essential importance to regulate mood and reduce anxiety [157,158]. Additionally, as previously mentioned, in newborn rats the exposure to music gave rise to increased BDNF concentration in the hippocampus, suggesting that such musical stimulation may have a beneficial potential effect with regards to neuroplasticity [140]. Although several of these studies with animal models have used music composed by Mozart, after inspiration from Rauscher et al.’s *Nature* study [142], it remains open to debate whether the effects are specific to this composer or this particular style of music, or whether they can be generalized to any music or even to any auditory stimulation with a periodic structure. In this respect, a recent study [137] again confirmed increased BDNF levels and improved learning performance in both rats and humans after listening to music by Mozart (12 h/day for 98 postnatal days in the case of rats, and 30 min/day for 6 days in the case of humans), in contrast to listening to retrograde Mozart music (keeping all the acoustic content but having a different rhythm) for the same amount of time. Hence, the authors put forward the hypothesis that the periodic rhythm lasting like a physiological cycle (20–30 s) might explain the neuroplastic and behavioral benefits from music listening. Further research is needed to explore this hypothesis [159,160].

In sum, a rather clear evidence on the association between BDNF and music exists, mainly showing the effect of music on the BDNF levels and expression. However, even though music has been widely used in recent research as a privileged tool to investigate the brain, the relationship between BDNF and music has not received the same attention that was dedicated to the one linking BDNF to sport, either in humans and animal models. Thus far, only two studies [137,161] have measured BDNF levels in humans after musical activities, compared with at least 29 studies showing BDNF modulation after sport practice (as reported by the meta-analysis dated back to 2015 [5]). On the other hand, genetic research demonstrates a relation between BDNF and music functions in humans with BDNF-related gene mutations associated with musical abilities [159], as well as increased BDNF microRNA transcription after a concert performance of professional ensemble musicians [162], or even simply after listening to 20 min of the violin concerto K. 216 by Mozart [160] (to note, these BDNF gene modulations were found in association with music functions, along with several other genes related to auditory system functions, affiliation, and neuronal development). Thus, future research is called for to better investigate this topic, and here we wish to provide some ideas that such research should pursue.

First and foremost, similar to what was done for sport research, cross-sectional studies could simply measure the plasma levels of BDNF in professional musicians compared to controls. A first study in this direction has been very recently performed by an Italian group, providing supportive evidence for higher plasma levels of BDNF in 21 musicians (with continuous practice of an instrument for over 5 years and having a musical degree) in contrast to 27 non-musicians [161] (matched for age, gender, and autism spectrum quotient). Future studies could explore a dose–response relationship between musical activities and BDNF levels, in order to understand whether the number of hours and years of training causally boost BDNF-mediated neuroplastic processes.

Furthermore, music listening, besides involving the drive to move (with dancing), is in and of itself a memory process: when we listen, we implicitly encode repeated melodic and rhythmic patterns, and are able to recognise them later on. In a recent study, we identified the systemic brain network changes in temporo-frontal brain areas, including the hippocampus, associated with these memory processes [163] (Figure 2). Future research could proceed in this direction and determine whether BDNF-related neuroplastic processes and the associated brain network changes could be triggered simply by passive exposure, namely by just listening to music without playing, as animal studies seem to indicate. Another relevant branch of research could consist of using music as a tool to explore functional neuroplasticity in connection to BDNF polymorphisms. In particular, a focus could be placed on the BDNF Val66Met polymorphism.

Along this line, investigating the relationship between such BDNF polymorphisms and memory-related brain responses in professional musicians, compared to controls, could provide further insights on the interplay between exercise-induced brain plasticity and BDNF.

## 6. Conclusions

In this paper, we reviewed evidence for the essential contribution of BDNF to brain biology and the impact of music and sport exercise on brain functions, paving the path for the hypothesis that these players are interrelated. Based on this and on established research on sport exercise, we propose that music (a special form of physical exercise) could be contemplated as a promising option for fostering BDNF production, ultimately improving human brain functioning. Notably, music is advantageous compared to physical exercise, since it is accessible even by patients with reduced mobility. In this context, studies establishing the relationship between dose and response would be crucial to understanding whether simple listening would suffice to produce neural preservation effects.

In summation, by providing evidence about the role that sport and music exert on neuronal functions, we propose the novel scenario that by boosting BDNF release in the brain, music, along with motor exercise, may become an inexpensive and safe strategy for preserving or restoring cognitive abilities in aged or demented individuals.

## Figures and Tables

**Figure 1 cells-10-00183-f001:**
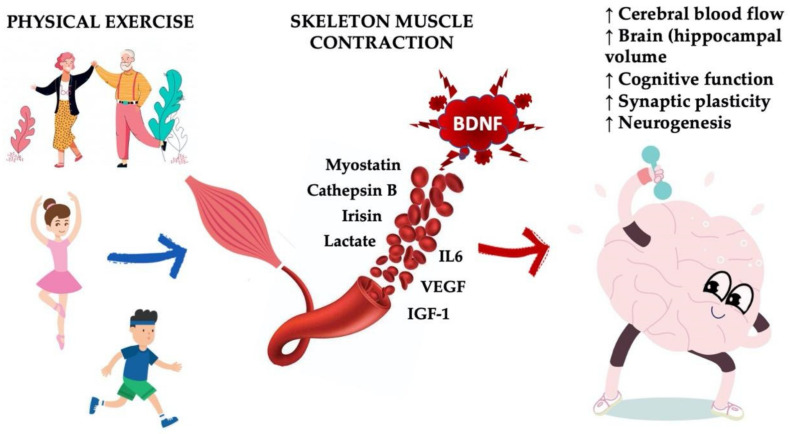
Proposed mechanism for the impact of physical activity on brain functions. The central nervous system (CNS) and peripheral skeletal muscle interconnection initiate movement, which results in skeletal muscle contractions and neuronal activation. Repeated physical activity triggers neurotrophic growth factors (e.g., brain-derived neurotrophic factor (BDNF), vascular endothelial growth factor (VEGF), insulin-like growth factor 1 (IGF-1), as well as myokine (e.g., irisin, cathepsin B) release; it also promotes neurogenesis, synaptic plasticity, and the improvement of cognitive performance.

**Figure 2 cells-10-00183-f002:**
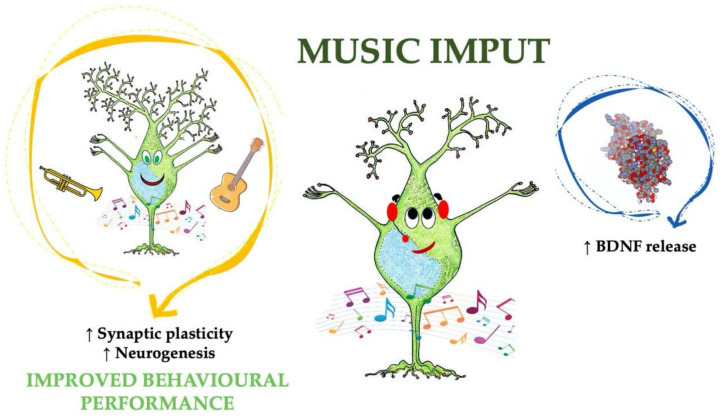
Schematic representation of the mechanisms leading to increased BDNF levels after music exposure and improved cognitive responses.

**Table 1 cells-10-00183-t001:** Cognitive and motor similarities and differences between sport and musical exercise.

Sport	Music
Similarities	
Moving body	Moving limbs and fingers, and occasionally the whole body
Watching activates action observation areas in experts	Listening activates action areas in experts
Watching sport activates reward brain circuits	Listening to upbeat music causes the drive to move and hence activates motor and reward areas
Behavioural studies show cognitive, mood, and health benefits	Behavioural studies show cognitive, mood, and health benefits
Differences	
Sport activities require body mobility	Musical activities do not require body mobility
Sport activities require awareness	Musical activities do not require awareness and can be proposed even to vegetative patients
BDNF studies on animals and humans	BDNF studies only on animals

## Data Availability

Not applicable.

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
