# Peer review of "Putting Cells in Motion: Advantages of Endogenous Boosting of BDNF Production"

_cells, 2021, doi:10.3390/cells10010183_

Round 1
Reviewer 1 Report
The article by Brattico and coworkers aims at reviewing the current evidence that BDNF is elevated in brain and serum following motor exercise and music. In particular, the paper intends highlighting the possible similarities between the effects on BDNF levels induced by sport and musical training, starting from the findings showing how listening to music rather than actual playing music, may also lead to increased BDNF levels. The authors put forward the hypothesis that music therapy may represent a valuable, non-invasive treatment for several brain diseases.
This is an interesting topic from the perspective of both clinical and basic research, however the manuscript presents several problems. In particular, there is an overwhelming number of wrong, missing or misplaced references, which is a very serious issue for a review article. In addition, the BDNF general description is inaccurate, and the paragraph on the relationship between music training/listening and BDNF does not consider some recent papers. As a whole, the paper fail to provide sufficient information concerning the parallelism between sport training and music training on brain plasticity and BDNF. Specific criticism are detailed below.
1) Introduction.
a) Reference [1] does not concern the topic indicated in the sentence: “Levels of BDNF are crucial in both normal and pathological aging and also in psychiatric diseases, in particular at the level of brain areas important in the codification of memory processes such as the hippocampus and parahippocampal areas”. But rather refers to the effects of physical exercise (PE) on BDNF levels. Other references should be used, for example: Ref. [31], [57], [58] and more recently, doi: 10.1007/s12035-018-1283-6; doi: 10.1007/s11920-017-0794-6.
b) The following sentence requires at least one reference: “Several lines of evidence from both animal models and physiological studies with humans point to the idea that motor exercise reliably boosts BDNF levels.”
2) BDNF:gene structure and protein localization.
a) The Neurotrophin 4/5 (not “4-5”, neither “4 and 5”, as erroneously written at pages 3 and 4) takes its name from the fact that it was first identified in Xenopus and named Neurotrophin-4 (NT-4) (doi: 10.1016/0896-6273(91)90180-8.). The homologue gene discovered in rodents, because of the evolutionary divergence from amphibians, was erroneously considered as a different molecule and initially named Neurotrophin-5 (doi: 10.1016/0896-6273(91)90287-a). However, when the mistake was recognized, it was too late to change the names and therefore this neurotrophic factor is now referred to as Neurotrophin 4/5 (doi: 10.1002/jnr.490320402; doi: 10.1523/JNEUROSCI.13-11-04961.1993).
b) The correct references for the sentence “Human, rat, and mouse BDNF are expressed from a single gene locus. Bdnf gene consists of eight 5′-untranslated exons and one protein coding 3′-exon all producing 24 different transcripts that are all translated into an identical mature dimeric protein highly conserved throughout mammals [21]” are, [23] for the human BDNF gene and Aid et al., 2007 for the rodent gene (doi: 10.1002/jnr.21139.). The correct number of 5’UTR exons for the human BDNF gene is 10. The correct number of BDNF transcripts is 22 for rodents (11 main transcripts having either a long or short 3’UTR), and 34 for the humans (17 main transcripts + 3’UTR short or long).
c) Ref [22] is the wrong reference, and it is not true that exons I-III are expressed predominantly in the brain, while exon IV has been found in the lung and heart. To see the correct tissue distribution of the different BDNF transcripts please, check the references by the Timmusk group mentioned above (Aid et al.,2007; Pruunsild et al.,2007).
d) The following sentence in incorrect, and requires additional references: “This complex Bdnf gene structure is believed to be essential for BDNF regulation at different levels through the interaction with a myriad of transcription regulatory factors or mRNA-targeting signals [23].” Actually, the transcription regulatory factors are not so many, and thus the term “myrad” appears inappropriate. Please, refer to the following two papers: Pruunsild et al., 2011 (doi: 10.1523/JNEUROSCI.4540-10.2011.); Kairisalo et al, 2009 (DOI: 10.1111/j.1460-9568.2009.06898.x). Instead, BDNF mRNA-targeting signals are indeed many, but no references are indicated. Please, find here some of the main references on this topic that should be cited or are already cited: Oe et al., 2016 (doi: 10.1247/csf.15015); Vicario et al., 2015 (doi: 10.3389/fnmol.2015.0006); Raju et al., 2011 (doi: 10.1091/mbc.E10-11-0904. Epub 2011 Apr 6.); Wu et al., 2011 (doi: 10.1111/j.1471-4159.2010.07166.x); Ma et al., 2010 = reference [50]; Chiaruttini et al., 2009 (doi: 10.1073/pnas.0902833106).
e) The reference for the following sentence is wrong: “Other factors such as physical exercise, seizures, ischemia, osmotic stress, and antidepressant treatment have also been involved in regulating Bdnf expression levels either at level of the promoter or by controlling BDNF translation modifications and/or stability [22].” The referred paper concerns BDNF regulation in post-mortem brains from AD and other types of dementia papers. The authors are invited to carefully chose the appropriate references.
f) The references for the following sentence are wrong: “Meanwhile, the intricate structure and regulation of Bdnf gene activity provide a broad susceptibility for the epigenetic control of Bdnf expression [24,25].” The cited papers (Sasi et al, 2017; and Deinhardt et al., 2014) do not refer to epigenetic control of BDNF expression. The authors should rather cite the following papers: Mallei et al., 2015 (doi: 10.1093/ijnp/pyv069.); Karpova, 2014 (10.1016/j.neuropharm.2013.04.002); Martínez-Levy GA, Cruz-Fuentes CS. Yale J Biol Med. 2014 Jun 6;87(2):173-86. eCollection 2014 Jun, and the already cited ref [61].
g) The references for the following sentence are wrong: “BDNF synthesis occurs in regions that participate in emotional and cognitive function, namely sensory cortices, hippocampus, amygdala, basal forebrain, dorsal vagal complex, hindbrain, and midbrain [24-26].” The three cited papers do not contain any information on anatomical distribution of BDNF nor on its role in emotional or higher cognitive functions, but rather refer to the different functions of mature and proBDNF in plasticity at single synapses (at pre- or post-synaptic level). The authors are invited to carefully chose the appropriate references.
h) The references for the following sentence are wrong: “From these areas BDNF can be retrogradely transported, thus reaching the cell bodies of the Raphe nuclei and locus coeruleus [27,28].” The words “Raphe nuclei” and “Locus coeruleus” do not appear in neither of the two papers. The authors are invited to carefully chose the appropriate references.
i) The reference for the following sentence is wrong: “BDNF is predominantly somatodendritic, but it is also present in the dendrites, and in close proximity to spines either in pre- or postsynaptic compartments [29].” In fact, the Altar et al., 1997 paper has become famous in the field because it was the first unequivocal demonstration of an anterograde BDNF transport in axons (not in dendrites). The paper to cite here, is another one, of the same year as that of Altar, namely Tongiorgi et al.,1997 (doi: 10.1523/JNEUROSCI.17-24-09492.1997), but the authors may simply use the already cited reference [30]. Instead, the authors should use reference [29] in addition to ref. [30-32] in the following sentence: “BDNF can undergo both retrograde and anterograde transport and this appears to be important for conferring the ability of locally translated BDNF to modulate synaptic transmission and synaptogenesis [30-32].”
j) From this point onward since the number of mistakes in citing the literature is overwhelming, I will only indicate the wrong references without or just limiting any further comments. The authors are invited to carefully chose the appropriate references. Wrong references which should be replaced are:
[28, 29];
[19, 30] – 19 is ok, but 30 is wrong;
[33 = this paper refers to music therapy to release anxiety in pregnant women, and does not provide any information supporting that: “The highest levels of pro-BDNF are observed perinatally, then it declines with age, although still detectable in adulthood [33]”;
[27, 39-41, in none of these papers the terms “glutamatergic stimulation, cerebral ischemia, hypoglycaemia” are present, these papers refer to the involvement of BDNF or NGF in the cellular mechanisms of AD];
[27,50] not the right citations for “Alzheimer’s disease (AD), Huntington’s disease and epilepsy”;
[22,36,49,51-56] not the right citations for “AD, Parkinson’s disease, amyotrophic lateral sclerosis, stroke and spinal cord injury” (only 22 is ok for PD, and 54-55 for AD).
The following sentence should be further discussed, because to my knowledge, the current literature, and even the cited paper express a more positive view on the FUS technique (either to enhance intranasal or intracortical delivery): “Even focused ultrasound has been proposed, but it is invasive and not always effective [65].
[33,66-69] of these papers, only ref [67] actually concerns stress effect on BDNF and its reversal by physical exercise.
[70] does not refer at all to “downhill running can induce time-dependent upregulation of skeletal muscle BDNF”, it is a paper on “ultrafast growth of high-quality uniform monolayer WSe2”. Why it is cited here?
3) Boosting endogenous BDNF: Music.
a) The authors state that “Previous studies have shown the effects of Mozart music on learning and memory and spatial-temporal tasks in both animals and humans [80-82]. In rats, a link between increased BDNF levels and exposure to Mozart’s music has been proposed [83].” However, this is a quite limited quote. The author should also cite and comment studies that failed to replicate the Mozart effect as well as cite those showing that similar effects can also be achieved with different types of music, thus challenging the concept of “Mozart effect” and expanding it to the idea that music therapy can help improving several types of cognitive performances in healthy or diseased brains. Some of these studies are already cited in the review, such as ref [82] and [88]. See for instance also the ref: DOI: 10.1038/srep18744.
b) Reference [83] does at all not concern: “Mozart’s music gave rise to increased BDNF concentration in the hippocampus, suggesting that such musical stimulation may have a beneficial potential effect with regards to neuroplasticity [83]”, instead it describes the development of the cerebellum.
c) What is the authors comment on the following question: are the therapeutic/cognitive effects involving BDNF production obtained by listening versus playing music same or different? Or in other terms are passive listening and active playing/singing having equivalent effects? Here is a non-exhaustive list of some reference not currently included in the review, which may be added and commented (additional ones comparing listening/playing or singing are welcome):
doi: 10.1007/s10072-020-04715-9
doi: 10.1111/j.1600-0404.2010.01417.x
doi: 10.1111/jgs.14361
doi: 10.1038/srep39707
DOI: 10.1080/15592294.2020.1809853
Author Response
√
- Introduction
Most sincere apologies for all the inaccuracies in the references. In this new version, we largely revised them, following reviewer suggestions.
- BDNF: gene structure and protein localization.
We reframed some sentences along the paragraph “BDNF: gene structure and protein localization” and added the appropriate literature in support of them. We extended the evidence supporting the role of BDNF in physiological and pathological brain conditions. All the changes are highlighted in red.
3) Boosting endogenous BDNF: Music.
Following reviewer suggestions, we critically discussed aspects related to the effects of music on brain functions also in respect of negative findings. We included two cartoons showing the mechanisms throughout physical exercise as well as music controls brain functions, figure 2 and 3.
References have been replaced as for reviewer suggestions. Moreover, we added all the new references recommended by the reviewer, including two studies of BDNF levels in humans after musical activities. One of them is from few months ago, so we could not notice it whereas the other skipped our attention when preparing the manuscript. Hence, we are truly grateful to the reviewer for pointing these studies at us. Furthermore, we appreciate the encouragement by the reviewer to further discuss the findings on the effects of music by Mozart or other composers on cognitive functions including BDNF-mediated neural plasticity. In the revised version of the manuscript, we also explicitly mentioned the relevance of defining the role of playing over mere listening for boosting BDNF effects.

Reviewer 2 Report
In this manuscript, the authors describe the effect of music and sport on BDNF release. The study has been submitted as a communication, although my understanding that it is a review which is very brief. I feel that a bit more comprehensive and detailed description would be helpful on the molecular mechanisms which are responsible for BDNF production. What is the reason why it is decreased in neurodegenerative disorders? Music and sport increases BDNF production through the same molecular pathways?
Author Response
We thank the reviewer for these suggestions. In this revised version, we discussed about the mechanisms responsible for BDNF functions and dysfunctions in the brain in more details. In addition, we included two cartoons showing the effect of physical education (Figure 1) and music (Figure 2) on brain functions.

Reviewer 3 Report
Brattico and colleagues review literature on BNDF. In particular, they focus on the actions of BNDF and the mechanism regulating BNDF expression, including exercise and music. Each of these areas are subjects of intense study so this review is timely and relevant to the field. Overall, the manuscript is clearly written, novel, topically relevant and of high quality. I feel that the manuscript a strong candidate for publication in Cells, but have a number of comments that if addressed should help to clarify and improve the manuscript.
Specific comments:
(1) Page 3, “In particular, BDNF is the second member of neurotrophic factor family to be identified, after NGF and before neurotrophin-3 and neurotrophins 4-5 [19,20].”
Please change to BDNF is the second member of neurotrophic factor family that has been identified, after NGF and before neurotrophin-3 and neurotrophins 4-5.
(2) Page 3, “Bdnf gene consists of eight 5′-untranslated exons and one protein coding 3′-exon all producing 24 different transcripts that are all translated into an identical mature dimeric protein highly conserved throughout mammals [21].”
Which species is referring to here? As human, rat and mouse have different number of BNDF transcripts.
(3) Page 3, “Each of these transcripts can be differently expressed in various tissues and Bdnf gene transcription is tightly regulated, cell-type specific, and controlled by neural activity. In particular, transcripts containing exons I–III are expressed predominantly in the brain, whereas exon IV has been found in the lung and heart [22].”
Ref. 22 has been mis-cited in page 3 and page 4. Please provide the correct reference.
(4) Page 4, “BDNF is synthesized in the brain but it can be secreted either by target neurons under physiological conditions or by astrocytes following injury or inflammation [22].”
What are target neurons? Reference is not correct here.
(5) Page 4, it would be really helpful if the authors could explain the difference between pro- and mature BDNF. For example, BNDF is initially synthesized as a precursor protein (pre pro-BNDF) that is cleaved into pro-BNDF, which can then be further cleaved into mature BDNF.
(6) Page 5, “Studies demonstrate that BDNF is necessary for LTP: high levels of BDNF facilitates LTP vs low levels diminish it. Moreover, the blockage of BDNF binding to TrkB nullifies LTP and neurogenesis [43].”
It is not clear to me what low levels of BDNF mean. Do the authors refer to knockdown or knockout of BNDF in animals? Also, the Ref. 43 is not the correct reference for this statement, please correct it.
(7) Page 5, “Moreover, it is known that NGF and BDNF contribute to survival of neurons and counteract neuronal degeneration and apoptosis in in vivo and in vitro animal models of AD”
It should be in vitro AD neuronal models.
(8) Page 6, “In the left hippocampus of rats BDNF was infused (0.5 μl/h) for 14 days, starting 7 days before the exposure to stressful conditions.”
What is the concentration of BDNF used here? For example, 1 mg/mL?
(9) Page 6, “The results of this study demonstrated that LTP, spatial learning and memory, usually compromised in chronic stress conditions, may be protected by the administration of BDNF [63].”
It would be more appropriate to use “could be” as it has been shown by previous study (Ref. 63).
(10) Page 6, it should be Table 2 not Table 1. There are some typos in the table. For example, Improves memory.
(11) Page 7, “For instance, downhill running can induce time-dependent upregulation of skeletal muscle BDNF [70], specifically an increase of post-exercise BDNF mRNA level 5d and 7d, BDNF protein level 1d, 3d, 7d and 14d, and augmented serum BDNF levels.”
The description is very confusing. Do the authors mean that BDNF mRNA levels increase 5 and 7 days after exercise? Also, Ref. 70 has been mis-cited. Please provide the correct reference.
(12) Page 8, “For instance, in mice it has been shown that exposure to music modulates neuroplasticity by increasing BDNF levels in the hippocampus [78] and hypothalamus [77].”
Ref. 77 and 78 need to be switched.
Author Response
We are grateful the referee comments and observations. Accordingly, all missed information underlined has been added in this revised version and all the related changes are marked in red.
English language suggestions have been included. Wrong references have been amended or switched following reviewer indications.

Round 2
Reviewer 2 Report
It is still not a communication but can be a comprehensive review.